# A Hybrid Genetic Algorithm Based on Imitation Learning for the Airport Gate Assignment Problem

**DOI:** 10.3390/e25040565

**Published:** 2023-03-25

**Authors:** Cong Ding, Jun Bi, Yongxing Wang

**Affiliations:** 1School of Traffic and Transportation, Beijing Jiaotong University, Beijing 100044, China; dingcong95@bjtu.edu.cn (C.D.); yxwang4@bjtu.edu.cn (Y.W.); 2Key Laboratory of Transport Industry of Big Data Application Technologies for Comprehensive Transport, Beijing Jiaotong University, Beijing 100044, China

**Keywords:** gate assignment, imitation learning, genetic algorithm, deep neural network

## Abstract

Airport gates are the main places for aircraft to receive ground services. With the increased number of flights, limited gate resources near to the terminal make the gate assignment work more complex. Traditional solution methods based on mathematical programming models and iterative algorithms are usually used to solve these static situations, lacking learning and real-time decision-making abilities. In this paper, a two-stage hybrid algorithm based on imitation learning and genetic algorithm (IL-GA) is proposed to solve the gate assignment problem. First of all, the problem is defined from a mathematical model to a Markov decision process (MDP), with the goal of maximizing the number of flights assigned to contact gates and the total gate preferences. In the first stage of the algorithm, a deep policy network is created to obtain the gate selection probability of each flight. This policy network is trained by imitating and learning the assignment trajectory data of human experts, and this process is offline. In the second stage of the algorithm, the policy network is used to generate a good initial population for the genetic algorithm to calculate the optimal solution for an online instance. The experimental results show that the genetic algorithm combined with imitation learning can greatly shorten the iterations and improve the population convergence speed. The flight rate allocated to the contact gates is 14.9% higher than the manual allocation result and 4% higher than the traditional genetic algorithm. Learning the expert assignment data also makes the allocation scheme more consistent with the preference of the airport, which is helpful for the practical application of the algorithm.

## 1. Introduction

Airport gates are the core resources in an airport operation process. Aircraft need to carry out various ground services (including passenger boarding, baggage handling, cleaning, and refueling, etc.) at a specific gate. The airport operation center has to allocate a reasonable gate for each arrival and departure flight, taking into account the flight arrival and departure time, aircraft type, and passenger and baggage information, as well as airport gate attributes, operation rules, and utilization status. It is a key task in airport operation management. With rapid growth in the number of flights, the complexity of gate scheduling work is increasing, and the shortage of parking space resources has been exposed. For many years, scholars have been constantly exploring the optimal assignment methods for gates to improve the utilization rate of parking gates, reduce airport operating costs, and improve passenger satisfaction. This problem is called the airport gate assignment problem (AGAP) [1].

The gate assignment problem is a combinatorial optimization issue, which is usually described through a mathematical programming model, that is, by optimizing one or more objectives under certain constraints. In order to simplify the problem, most of the existing studies only consider the hard constraints of the airport, such as aircraft-type restrictions; two consecutive flights allocated to the same gate cannot have conflicts regarding slot occupancy time. The algorithms for solving the gate assignment model mainly include two categories: exact algorithms [2] and heuristic algorithms [3]. This method may be useful for certain new airports to achieve specific goals. However, in airports that have been operating for a long time, there are often many unique and uncertain soft constraint rules. Some rules are difficult to describe quantitatively. For example, airports usually assign flights of the same airline to a specific area, assign flights with long parking time to remote gates, or even assign flights to gates near to the runway to reduce taxi fuel consumption. These soft constraints indicate that the airport has formed specific allocation preferences for flights with different attributes through the process of long-term operation. If the algorithm does not learn these preferences, the gate allocation results will not conform to the daily allocation habits of the airport, and the optimization results still need to be adjusted manually. Therefore, at this stage, a more intelligent method would be to add a model that is able to learn human-assigned results to predict and recommend the common gates similar to those of the allocators, which are more acceptable. In recent years, with the rapid development of artificial intelligence, the remarkable achievements of deep learning and reinforcement learning in the fields of robotics have demonstrated strong learning ability and sequential decision-making ability. Deep neural networks can automatically learn the characteristics of the target, unlike in manual algorithm design. Among the different types, imitation learning (IL) is a supervised learning method that extracts useful knowledge from expert decision datasets and reproduces their behavior strategies in similar environments. IL has made important contributions to driving behavior simulations [4], robot control [5], navigation tasks [6], and other fields. These new methods have the advantages of fast solving speeds and strong model generalization abilities, which provide a new method for solving combinatorial optimization problems.

According to the above considerations, this paper creatively proposes a hybrid genetic algorithm based on imitation learning to solve the gate assignment problem. The presented work can be summarized as follows:We transform the mathematical model of the gate assignment problem to a Markov decision process. The state space, action space, rewards, and state transition of the problem have been defined.We propose a hybrid genetic algorithm based on imitation learning (IL-GA) to solve the gate assignment problem. This algorithm includes two stages: In the first stage, a deep policy network is designed to learn the decision-making process of human allocation experts. The network is trained by historical flight assignment trajectories and outputs the gate selection probability of flights. Considering that it is impossible to obtain a better solution only by learning the decision-making trajectory of human experts, a genetic algorithm is used in the second stage in order to optimize the number of flights assigned to gates. At the same time, the optimal solution can be provided to the policy network as dataset aggregation for further training.We use real airport data from Lijiang Airport in Yunnan, China, to verify the rationality and effectiveness of the model and algorithm. The experimental results show that the genetic algorithm combined with imitation learning can not only better meet the airport gate allocation preference but also help to provide a good initial population, shorten the iteration time, and accelerate the convergence of the algorithm to better solution.

The rest of this paper is organized as follows: the Section 2 introduces a literature review of the airport gate assignment problem; in the Section 3, the mathematical model and the transformation of the Markov decision process for the gate assignment problem are provided; the Section 4 introduces the steps of our hybrid genetic algorithm based on imitation learning; the Section 5 introduces the experiment and results; and the Section 6 provides the conclusion and outlook.

## 2. Literature Review

With the development of the air transport industry, the amount of air routes, flights, and passenger traffic is growing, which makes airport scheduling and the management of gate resources more complex. Especially for large hub airports, improper scheduling may cause flight delays, reduce passenger satisfaction, and increase airport operating costs. There are generally two ways to solve the shortage of airport gate resources: one is to directly expand the airport and add gate service facilities. However, this option requires a lot of money, time, and manpower. Therefore, a more practical method is to effectively use the existing gate resources, optimize the gate allocation scheme, and improve the speed and effect of the gate assignment through algorithms.

Some early studies adopted expert knowledge systems [7,8,9] to solve the problem of gate assignment. The system builds a knowledge base and inference process according to the allocation rules to help complete the gate allocation task. However, the result of this method is only one feasible solution. Different goals cannot be completely understood and optimized through rules. The knowledge base also needs to be constantly updated and maintained alongside the development of the airport. Therefore, much of the literature chooses to use mathematical programming for research.

Early research on the mathematical models for the gate assignment problem mainly focused on the optimization of a single target, for which many studies considered the travel distance and time of passengers. For example, Babic et al. [10] minimized the average walking distance of arriving and leaving passengers. Yan and Tang [11] minimized the total waiting time of passengers from the apron to the boarding gate. Kim et al. [12] minimized the transfer time of passengers in the terminal. From the airport perspective, some studies aim to improve the flight rate assigned to the gate near the terminal [13,14] and the robustness of the gate allocation plan [15,16]. In addition, there are certain considerations regarding the goals of airlines. Maharjan and Matis [17] established a gate network traffic optimization model to minimize the fuel consumption cost of aircraft taxiing. Xiao et al. [18] minimized the total cost of passenger transfer, flight accommodation, and gate occupancy. In recent years, with the complexity of airport business, it has become difficult for single-objective optimization to correspond to the actual situation, so many multi-objective gate assignment models have now been designed. Zhang and Klabjan [19] considered multiple objectives to minimize the total number of flight delays, the number of gate reassignment operations, and the number of missed passenger transfers at the same time. They used the weighted sum method of multi-objective optimization to solve their model. Aktel et al. [20] considered minimizing both the number of non-boarding flights and the total walking distance. In the study by Behrends and Usher [21], the first goal was to minimize the exit delay caused by cargo passing through the terminal, while the second goal was to minimize the delay caused by aircraft passing through the airport.

In addition to the optimization models, the other main research angle is the algorithms used to solve the gate assignment problem. Common algorithms can be divided into exact algorithms and heuristic algorithms. For the exact algorithm, Bi et al. [22] used the branch and price algorithm to solve the model with the goal of maximizing the number of passengers using jetways. They added a certain probability of heuristic search to each search node so that the solution speed was improved. However, the gate assignment problem has proven to be a hard NP problem, and the exact algorithm cannot be applied to large occasions, so more research focuses on heuristic algorithms. Seker and Noyan [23] studied the gate allocation under the uncertainty of flight takeoff and landing time, and they established a stochastic programming model, which was solved by an improved tabu search algorithm. Yu [24] focused on passenger transit satisfaction in real-time distribution and solved it with heuristics. Deng et al. [25] used improved particle swarm optimization (PSO) to solve the problem, taking into account the walking distance of passengers, free time at gates, and the utilization rate of gate space. Deng et al. [26] also proposed an ant colony optimization (ACO) algorithm for solving their multi-objective model. Stollenwerk et al. [27] considered the transfer time of passengers and adopted an improved quantum annealing method to solve it. Aoun and EI Afia [28] solved the problem of aircraft position assignment under uncertain conditions based on the Markov decision process and used random parameters to represent fluctuations in flight operations. Another paper by these authors [29] took into account that aircraft could be delayed, established an allocation model based on a multi-agent Markov decision-making process, and expressed the stand as a collaborative agent trying to complete a series of flight assignment tasks, finally providing the staff with a robust priority solution. In 2018, they [30] also designed a time-varying multi-agent Markov decision process model for other random situations in aircraft slot allocation, which provides allocation schemes in each time series.

The abovementioned research is mainly based on macro objectives and solved by iterative algorithms. Little research has been conducted which analyzes the historical allocation data of the airport and considers the existing allocation mode and preference. In this way, even if an optimal solution can be obtained, the actual operation of the airport may not be satisfied. Therefore, the hybrid genetic algorithm based on imitation learning (IL-GA) proposed in this paper aims to analyze and learn the existing allocation preferences of the airport before the gate allocation and then optimize on this basis, which is of great significance to the practical application of the algorithm.

## 3. Model Construction

In the actual operation process of the airport, the gate allocation work includes two stages: pre-allocation [31] and real-time allocation [32]. Pre-allocation refers to the assignment of an appropriate gate for all flights in advance according to the flight plan of the next day. Real-time allocation is the modification of the pre-gate allocation plan when the slot occupancy time conflicts due to flight delays or other uncertainties. A good pre-allocation plan should be able to summarize experience from the historical real-time allocation process, such as allocating flights that are easy to delay or have a long parking time to remote gates, in advance. In this way, the pre-allocation plan can well tolerate the actual changes in the airport, enabling maximization of the stability of the entire airport operation system and reductions in the workload of secondary assigned tasks. Therefore, this paper mainly examines the optimization of the pre-allocation scheme of airport gates by learning the experience from historical allocation data. In this section, we first give the mathematical model description of the gate assignment problem and then use the Markov decision process to transform the model.

### 3.1. Mathematical Model of AGAP

The related research objects and concepts of the airport gate assignment problem involved in this paper are described as follows:Flight pair

An aircraft performs an inbound flight mission and arrives at the airport. After a period of stay, it performs another outbound flight mission and leaves the airport. A paired flight only needs one specific gate for parking. The attributes of a flight pair include arrival time, departure time, aircraft type, international/domestic type, VIP type, overnight type, etc.

2.Contact gates

These are gates located near to the terminal. Passengers can board and depart through a bridge connecting the aircraft and the boarding gate.

3.Remote gates

These are gates located at the apron, far from the terminal. Passengers have to take a ferry vehicle to connect to the terminal.

4.Gate preference

Each paired flight has a potential gate selection preference, which depends on its attributes. For example, the airport might prefer to park VIP flight pairs at the contact gates to facilitate important passengers, but an overnight flight pair may tend to be assigned to remote gates.

According to the above description, parameters and decision variables in the model are defined, as shown in Table 1.

#### 3.1.1. Constraints

The gate assignment problem has multiple constraints. Due to the different actual conditions and operational decisions of each airport, the constraint rules used in the gate assignment process are also different. The constraints used in this paper are as follows:One aircraft (flight pair) can only park at one gate.(1)$$\sum_{k\in G}x_{ik}=1,\forall i\in F,k\in G$$The aircraft type and international/domestic attribute must be within the allowable range of gates.(2)$$x_{ik}=0,\forall i\in F,k\in G,f_{i}^{T}\notin g_{k}^{T},f_{i}^{N}\neq g_{k}^{N}$$Two consecutive aircraft assigned to the same gate should add a necessary safety time interval between them. For the convenience of calculation, the conflict relation matrix Q of any two flights is calculated in advance according to the arrival and departure times. If the interval between flight i and flight j is less than the minimum safety time interval $T_{buffer}$, $Q_{ij}=1$; otherwise, it is 0. Then, this constraint can be expressed as follows:(3)$$x_{ik}+x_{jk}\leq 1,\forall i,j\in F,k\in G,Q_{ij}=1$$Two flights assigned to adjacent gates cannot arrive or leave at the same time. Because the distance between adjacent gates is close, sliding in or pushing out at the same time can easily cause wing scratching. Similarly, a conflict matrix W is generated in advance to indicate that the arrival or departure time of two flights does not meet the safety interval $T_{neighbor}$. Then, the constraint can be expressed as Equation (6):(4)$$x_{ik}+x_{jl}\leq 1,\forall i,j\in F,k,l\in G,|k-l|=1,W_{ij}=1$$

#### 3.1.2. Objective Function

Maximize the number of flight pairs assigned to contact gates

Contact gate resources are preferred by most aircraft. After the aircraft stops at the contact gate, not only can passengers arrive at the terminal quickly but it is also more convenient for the aircraft to check, refuel, add water, and other services. For airports, assigning flights to contact gates can also reduce the use of ferry vehicles, reduce the complexity of ground traffic in the airfield, and reduce the workload of ground support personnel. Therefore, maximizing the utilization rate of the contact gates is often the core of evaluating the gate allocation scheme of large hub airports. The objective function to maximize the flight pair ratio assigned to contact gates is described in Equation (5).
(5)$$F_{1}=\max\sum_{i=1}\sum_{k=1}x_{ik}g_{k}^{B}$$

2.Maximize the total gate preferences

Although contact gates are more convenient for flights, their number is limited. Ultimately, some flights must be assigned to the apron. Therefore, it is necessary to consider which flights are more suitable for allocation to contact gates, and this depends on the attributes of the flights. According to our investigation at Lijiang Airport, there are priority rules for assigning flights to contact gates; for example, international flights have higher priority than domestic flights, flights with short parking times have higher priority than those with long parking times, and so on. Only increasing the contact gate utilization rate would result in certain high-priority flights being assigned to the far apron, which is unacceptable. Therefore, a gate preference, $p_{ik}$, for each flight is defined in this paper, which should be clearly given by the airport managers based on the actual priority rules. The objective function to maximize the total gate preferences is described in Equation (6).
(6)$$F_{2}=\max\sum_{i=1}\sum_{k=1}x_{ik}p_{ik}$$

However, assigning an appropriate value to the gate preference of each flight is a complex task, which requires one to consider various attributes of the flights and establish the inference process. In fact, the information of gate preference is reflected in the historical allocation data of airport operations. Therefore, this paper chooses to use the method of imitation learning to fit the expert assignment trajectory data to replace the process of assigning values to this parameter. Imitation learning needs to define the problem as a sequential decision problem and find out the state and action to be learned, so the Markov decision process is used to redefine the gate assignment problem in Section 3.2.

### 3.2. Markov Decision Process of AGAP

The mechanism of the Markov decision process is shown in Figure 1. At time t, an agent obtains the observation $o_{t}$ through the environment state $s_{t}$. The agent (generally called policy π) decides to take action $a_{t}$ according to the $o_{t}$. After the action $a_{t}$ is executed, the observation of the environment state at the next moment becomes $o_{t+1}$, and then the policy π tells the agent to take action $a_{t+1}$, until the end of the game. This process is very similar to gate assignment work. The airport gate allocation personnel generally assign the flight to the gate according to a sequence. That is, for a flight list $\left[f_{1},f_{2},\ldots ,f_{n}\right]$ to be allocated, the dispatcher first assigns a suitable gate to flight $f_{1}$ according to the gate occupancy status of the airport at that moment and the information of flight $f_{1}$. After that, the gate occupancy status changes, and flight $f_{2}$ selects a gate according to the changed status until all the flights are assigned. Therefore, the process of gate allocation can be regarded as a sequential decision-making problem with a Markov property. The objective of the Markov decision process of gate assignment problem is to find an optimal policy $\pi^{*}$, so that the agent can allocate flights in order according to this policy and finally obtain the maximum benefit. A Markov decision process has four elements: state space, action space, reward, and state transition [33]. The four elements of the Markov decision process model for the gate assignment problem are described as follows:

State Space

The state refers to the external information that an agent can observe when making decisions. For the gate assignment problem, waiting flights are first arranged into a sequence (usually in the order of departure time). When assigning the first flight, the agent can observe the attributes of the current flight and the use status of each airport gate. Figure 2 shows an example of the state representation. The state is represented by a fixed-length code and is divided into three parts:(7)$$o_{t}=[o_{fi},o_{ti},o_{gi}]$$

$o_{fi}$ is the coding set of flight attributes, including aircraft type, international/domestic type, VIP type, and overnight type. We use the one-hot code method [34] to represent the attributes of flights and then splice them together.$o_{ti}$ is the gate occupancy time of the flight. We divide 24 h into 288 bits every 5 min. The bits between the arrival and departure time of the flight are set to 1; otherwise, it is 0.$o_{gi}$ is the coding set of gate status for a current flight. We use the constraints to filter out the available gates and assign 1 to indicate availability; otherwise, 0 is assigned.

2.Action Space

Action $a_{t}$ is the parking gate allocated to the first flight based on the state $o_{t}$. Therefore, the size of action space A should be equal to the number of gates set G:(8)$$a_{t}\in A=G=[0,1,2,\ldots ,k]$$

$a_{t}=k$ means that gate k is chosen by the policy π to be allocated to the current flight.

3.Reward

After each flight is allocated, the agent will receive an instant reward. According to objective Functions (5) and (6), the single-step reward is defined as whether the action is a contact gate and the flight’s preference for this gate. The total reward is expressed by R, and our goal is to find the best policy to obtain the maximum total reward.
(9)$$R=\sum_{i=1}\sum_{k=1}(x_{ik}g_{k}^{B}+x_{ik}p_{ik})$$

4.State Transition

After a flight is allocated, the status of the environment changes to the next flight, including flight attributes and gate occupancy time. The result of the previous flight shall be added to the use status of the gate, indicating that this gate has been occupied. This paper does not consider the random delay of flights, so the probability of state transition is set to 1 according to the current state and action. When the last flight is assigned, the whole process ends.

## 4. A Hybrid Genetic Algorithm Based on Imitation Learning for AGAP

For the mathematical model in Section 3.1, we can use metaheuristic algorithms to find the optimal solution, such as genetic algorithm [35]. However, heuristic algorithms usually start from a random solution. When faced with new scenarios, the whole algorithm needs to be rerun. Often, the optimal results are unstable and very dependent on the quality of the initial solution. For the Markov model of gate assignment proposed in Section 3.2, our goal is to train a good policy, so that we can obtain a good allocation scheme by making decisions in sequence based on this policy. At present, certain reinforcement learning algorithms [36,37] are used for training the policy. However, this method based on cumulative rewards has a huge search space. Only when a lot of training data are obtained after a long time of exploration can better performance be achieved. In addition, the reward of the gate preference part in this paper cannot easily offer quantitative values, so the reinforcement learning method is not suitable. In contrast, imitation learning can solve such problems more quickly. Imitation learning refers to learning directly from the example provided by the experts. That is, learning the optimal decision trajectory directly can greatly reduce the invalid data in training. Therefore, this paper proposes a two-stage hybrid algorithm for the gate assignment problem, which combines the generalization ability of imitation learning and the optimization ability of the genetic algorithm.

### 4.1. General Scheme

The schematic diagram of the hybrid algorithm is shown in Figure 3. In the first stage of the algorithm, we use the mechanism of imitation learning to train a policy network. A large amount of manual gate assignment trajectory data are prepared to generate the training samples and tags. The policy network is trained using the supervised learning method. This stage is offline, aiming to allow the network to make human-like choices when allocating gates. The trained policy network can provide the probability distribution of the gates after observing the gate status and flight attributes. That is, the policy network knows which gates are more likely to be selected for a flight; otherwise, it is impossible for this flight to be allocated to gates with less probability in reality. In the second stage, we aim to quickly solve an online gate allocation instance. A good initial population is first generated according to the policy network and *ε*-greedy principle. The crossover, selection, and mutation operators in the genetic algorithm are used to optimize the population and seek the optimal solution of the problem. Finally, in order to make the policy network more intelligent, a large number of optimal solution data from the genetic algorithm are collected as data aggregation to further train the network. The details of each stage in the algorithm are introduced as follows.

### 4.2. Imitation Learning Stage

Imitation learning refers to learning from examples provided by a teacher. Generally, the decision data of human experts $\left\{\tau_{1},\tau_{2},\tau_{3},\ldots ,\tau_{n}\right\}$ are collected, and each decision contains a sequence of states and actions $\tau_{i}=[o_{1}^{i},a_{1}^{i},o_{2}^{i},a_{2}^{i},\ldots ,o_{m}^{i},a_{m}^{i}]$. After extracting all the “state and action pairs” to construct a new set $D=\{\left(s_{1},a_{1}\right),\left(s_{2},a_{2}\right),\ldots ,(s_{n},a_{n})\}$, we are able to learn a classification (for discrete actions) or a regression (for continuous actions) model by taking the states as features and the actions as labels. The training objective of the model is to match the state action trajectory distribution generated by the model with the input trajectory distribution.

For the problem of gate assignment, a large amount of data accumulated during airport operation can be regarded as an expert example. In this paper, a deep neural network is constructed to learn the human trajectory data. The structure of the network is shown in Figure 4. The input to the network is the state space described in Section 3.2, that is, the flight and gate information that can be observed by the agent during allocation. The output dimension of the network is consistent with the number of gates. The probability of each gate being selected is output through normalization processing. Finally, the allocation actions can be obtained according to the probability.

The training goal of the policy network is to maximize the gate preference, so the cross-entropy error function [38] is used as the optimization goal to optimize the model, as shown in Equation (10).
(10)$$L\left(y,a\right)=-\frac{1}{N}\sum_{n\in N}y_{n}loga_{n}$$
where N is the number of training samples, $y_{n}$ vector is the label of samples, and $a_{n}$ vector is the output of the network. We convert the manually allocated gate into a one-hot code, that is, a code with the same length as the number of gates. Only the allocated gates are set to 1, and the others are 0. For example, if the number of gates is 4, $y_{n}=[0,1,0,0]$ indicates that the flight is assigned to gate 2 in the sample and $a_{n}=[0.24,0.62,0.08,0.06]$ indicates the output probability of the network. If the network is well-trained, the probability of it choosing a gate in a human-like way is larger, thus obtaining a similar result, so as to meet the gate preference of the flight.

### 4.3. Genetic Algorithm Stage

Although the trained policy network can be used to quickly assign a gate, its goal is only to meet the gate preference, and it cannot achieve the optimization of the contact gate leaning rate. Therefore, we also need to use a genetic algorithm for optimization in the second stage.

The basic idea of the genetic algorithm is to optimize a population of possible potential solution sets of the problem. The population consists of a certain number of individuals, each of which is actually an entity with chromosome characteristics. After the emergence of the initial population, according to the principle of survival of the fittest, the evolution of generations produces better and better individuals. In each generation, individuals are selected according to their fitness, and a new population is generated through two genetic operators: crossover, and mutation. The whole process is similar to natural evolution, and the optimal individual in the last generation of a population can be decoded as the approximate optimal solution of the problem.

Considering the characteristics of the gate assignment problem, the key steps of the genetic algorithm are designed as follows:Chromosome coding

Aircraft (flight pairs) and gates are involved in the gate assignment problem, so each chromosome needs to reflect the combination of these two resources. In this article, we use real number encoding, as shown in Figure 5, and the length of chromosomes is the total number of flight pairs; that is, the length of chromosomes is determined by the number of flights. The gene bit on the chromosome represents the assigned gate for the aircraft executing the flight pair, and the number on the gene bit represents the number of the assigned gate. For example, flight 1 is assigned to gate 5, flight 2 is assigned to gate 9, and so on.

2.Generation of the initial population

The genetic algorithm takes an initial population containing multiple chromosomes as the starting point of iteration, and the selection of initial population affects the convergence efficiency and search quality of the algorithm. Hub airports generally have a large number of gates. If the initial population is generated randomly, the search space will be huge, and there will be a lot of invalid search actions. The policy network trained in the first stage can represent the aircraft’s gate preference, so selecting the gate with a high probability for each flight to search can greatly improve the efficiency of optimization. In this paper, we use the policy network combined with the *ε*-greedy criterion to generate the initial population. The *ε*-greedy criterion refers to the compromise between exploration and utilization based on probability. After the policy network provides the flight’s preference for each gate, in order to ensure the diversity of the population, we still randomly select a gate for it with a probability of *ε*, or select the gate with the highest preference for the flight with a probability of 1-*ε* (if the preference is the same, select one at random). The steps are summarized in Algorithm 1.
**Algorithm 1:** Generation of the initial population for AGAP**Input:** An instance of flights F=[f1,f2,f3,…,fi] waiting for assignment;     Gate list G=[g1,g2,g3,…,gk];    Policy network π;    Greedy probability ε;    Population size N.**Process:**pop=[ ]//The set for population**for** t=1,2, … ,N **do**  chrom={}//A single chromosome  **for** f1,f2,f3,…,fi **do**    pik=π(ofi,oti,ogi)//Calculate gate selection probability    **if** rand()<ε **then**      Randomly add a gate to chrom according to pik    **else**      Add a gate with max⁡pik to chrom    **end if**
  **end for****end for****Output:** Initial population pop


3.Selection, crossover, and mutation operators

After the initial population is generated, we need to use the selection, crossover, and mutation operators to update the population through iteration. First of all, excellent individuals should be screened according to the fitness of chromosomes. The fitness function used to evaluate the population is the same as our optimization objective, that is, the number of flights allocated to the contact gate, as shown in Equation (11):(11)$$fitness=\sum_{l\in L}g_{l}^{B}$$
where l is the gene locus and L is the chromosome length. $g_{l}^{B}$ means that if the gate on gene locus l is a contact gate, its value is 1; otherwise, it is 0.

In this study, the most commonly used roulette wheel method [39] is utilized for selection. It is a proportion-based selection, which uses the proportion of individual fitness to determine the possibility of its retention. If the fitness of a body i is $f_{i}$ and the population size is N, the probability of its selection is expressed as:(12)$$p_{i}=\frac{f_{i}}{\sum_{i\in N}f_{i}}$$

For the crossover operator, the individuals in the population are paired first. Then, according to the crossover probability of contemporary individuals, whether to conduct a crossover operation is determined. For chromosomes requiring a crossover operation, two crossover points of the crossover operation are randomly generated. Assume that the selected intersection is at points 1 and 2. Finally, two random intersections (point 1 and point 2) exchange their gates. The process of the crossover operator is shown in Figure 6.

The crossover operation will make the originally feasible chromosomes unfeasible. Therefore, during the mutation process, the unfeasible chromosomes will be transformed into feasible chromosomes through gene mutation. The process of mutation is as follows:

Step 1: Traverse the chromosomes in the population. For the current chromosome, obtain the flights and the gates according to the sequence of chromosome coding. If the flight stops at the gate meeting the international and domestic attribute constraints, gate-type constraints, same gate conflict constraints, and adjacent gate conflict constraints, the chromosome will still be retained. Otherwise, a random available gate will be reassigned to the flight.

Step 2: After all flights of the current chromosome are judged, they will be added to the next population. If the set of feasible gates for general travel is empty, the chromosome will be discarded until all chromosomes in the population are traversed.

Step 3: If the number of chromosomes in the final next population list is less than the setting size, randomly select chromosomes from the new population to copy until the set population number is reached.

### 4.4. Data Aggregation

The policy network in the first stage of the algorithm proposed above is obtained by learning the allocation trajectory of human experts. Due to the inevitable errors in the training process, the trained network will gradually accumulate errors when making multi-step decisions, and the final results may not be ideal. Therefore, this paper uses the method from study [40] for reference, adds an additional data aggregation link, mixes the optimal solution dataset obtained by the second stage genetic algorithm with the original dataset, and then trains the network again to make it more intelligent. The steps of data aggregation are as follows:
Training the policy network $\pi_{\theta }$ through the human allocation trajectory dataset D;Using $\pi_{\theta }$ to solve new instances and save assignment results as dataset $D_{\pi }$;Optimizing the result of each instance in $D_{\pi }$ through genetic algorithm;Data aggregation: $D\leftarrow D\cup D_{\pi }$;Return to the first step.

## 5. Experiment

The Markov decision model of the gate assignment problem and a two-stage hybrid genetic algorithm based on imitation learning are presented in the previous sections. In this section, we use the real data of an airport to verify the rationality and effectiveness of the model and algorithm.

### 5.1. Experimental Data and Analysis

The data in this paper come from Lijiang Airport in Yunnan, China, which is a hub airport mainly serving tourism. The airport gates are distributed as shown in Figure 7, comprising 28 gates, including 10 contact gates and 18 remote gates. The attributes of each gate, such as the aircraft type and international/domestic attributes, are shown in Table 2. Among them, gate 3 and gate 5 are special gates for international flights, and gate 7 is only available for the B737 aircraft type. We obtained 14,000 actual flight gate assignment trajectory data from the airport from December 2020 to June 2021. This dataset represents about 180 days of flight data, and each record is a flight pair. Partial flight pairs are shown in Table 3 as samples. These are the assignment results from manual experience. Statistics and the analysis of the manual assignment trajectories can help us understand the gate assignment preferences of the airport and can also be used to train and verify our policy network.

By analyzing the data, it can be seen that the range of gates selected for each flight can be greatly reduced according to their attributes. We found that in addition to the hard security constraints, the airport is more sensitive to the occupancy time of flights when selecting gates. We calculated the average number of flights assigned to each gate in a day according to flight occupancy time. Figure 8 shows the influence of flights with different occupancy times on the selection preferences of gates. For flights with a short occupancy time (less than 2 h), the airport mainly chooses the contact gates and prefers to use gates 15, 16, 11, 10, and 9. For the flights with a medium occupancy time (2~3 h), the other contact gates and remote gates near the terminal will be selected, such as 12, 13, and 14. For flights with long occupancy time (more than 3 h or overnight), whether the flight arrives during peak hours or not, the airport will prioritize the remote apron, i.e., gates such as 17~19/20~28. Considering the gate preferences of flights can not only help to make the gate allocation more reasonable but can also reduce the search scope during optimization, we used a deep policy network to imitate and learn the gate preferences of flights.

### 5.2. Policy Network Construction and Imitation Learning

In the first imitation learning stage of the algorithm, the flight attributes selected in this paper include flight VIP type, overnight type, international/domestic type, and the gate occupancy time. The initial status of the airport gates is empty; that is, the first flight can be assigned to any available gate. After the first flight is assigned, the status of the second flight is transferred to the new status. This process is repeated until each flight in the dataset can be coded into the state space, as described in Section 3.2, and then the input of the policy network is determined. The output of the policy network is the selection probability of each gate, so the number of units in the output layer is set to 28 and the output is transformed into a probability vector by the Softmax layer. We chose a deep full-connection neural network (FC) to build the policy network. The settings and training parameters involved in the network are shown in Table 4 and Table 5. We chose 80% of the flight data for training and 20% for testing. The training process in this paper is implemented with Python 3 and Pytorch, the integrated development tool is PyCharm, and the running environment is an Intel (R) Core (TM) i7-7700HQ CPU @ 2.80 GHz with 16.00 GB of RAM. The iterative process of cross-entropy loss in the training epochs is shown in Figure 9.

It can be seen from Figure 9 that the error decreases rapidly in the first 50 epochs of training and then converges gradually. After training, we use the policy network to verify the testing data. We output the probability of selecting gates for each flight and rank them from high to low. Due to the inevitable error, it is too harsh to judge the accuracy of the network only by selecting the gate with the highest probability. Therefore, the top five gates with the highest probability for flights are selected as the predictive gate results. If the actual results are among the top five predictive gates, the model predictive result is considered reasonable. Table 6 gives some examples of comparisons between the predictive gates and actual gates of flights. According to the test result statistics of 2800 flights, the gates with maximum probability in the predictive results of 1206 flights are consistent with the actual gate, the top five gates in the prediction results of 2328 flights include the actual gate, and the top ten gates in the prediction results of 2685 flights include the actual gate. The statistical test results show that 83% of the actual allocation results of flights are within the top five of the predictive results, indicating that the network prediction is more accurate and can accurately locate the flight’s gate preferences.

### 5.3. Genetic Algorithm Stage

Although the policy network can help us better fit the gate preference of flights and reduce the range of gate selection, it cannot maximize the number of flights assigned to contact gates when directly using it to allocate a large number of flights. Therefore, according to Algorithm 1 proposed in Section 4.3, we use the policy network and the ε-greedy principle to generate the initial population and then use the genetic algorithm to optimize the number of flights assigned to contact gates. The relevant parameter settings of the genetic algorithm are shown in Table 7. Of these, the number of gates with high preference selected through the policy network and the *ε*-greedy parameter are variable, which are used to test the sensitivity of the algorithm.

We designed eight different instances to test the optimization effect of the algorithm. Each instance involves a set of flights waiting to be assigned on a certain day and has different parameters. In order to reflect the optimization effect of the algorithm proposed in this paper in the iterative process, we chose the basic genetic algorithm and Gurobi optimization solver for comparison. The initial population in the basic genetic algorithm is randomly generated. The comparison of the iterative process is shown in Figure 10, and the detailed parameter settings and optimal result statistics of the instance are shown in Table 8.

According to the results of each instance in Table 8, it can be seen that the direct use of the policy network trained by manual data is only able to allocate 67.7% of flights to the contact gates, which does not improve the utilization rate of the contact gates. However, using the policy network to generate the initial population and iterating on this basis can help the genetic algorithm to better optimize, allocating 81.5% of flights to contact gates. Compared with the basic genetic algorithm, which generates initial solutions randomly, the flight rate allocated to contact gates is increased by 4%, and the optimization result is closer to the optimal solution obtained by using the Gurobi optimization solver. This proves that the two-stage hybrid genetic algorithm based on imitation learning (IL-GA) proposed in this paper is effective.

In addition to the optimization effect, it is necessary to analyze the impact of parameters *ε* and *GP* on the algorithm iteration process. *GP* refers to the number of gates with higher gate preference provided by the policy network, that is, the range of gates selected by each flight when generating the initial population. *ε*-greedy represents the probability of not selecting the gates provided by the policy network but randomly selecting a gate. It can be seen from Figure 10a–c that when *ε* is small, the iteration number of the genetic algorithm incorporating the policy network is greatly reduced, and it can basically converge to the optimal solution within 50 generations, which shows that the gates recommended by the policy network are reasonable. When *ε* is big, too many randomly selected aircraft positions will lead to slow convergence and deviation from the optimal solution. Meanwhile, there are also errors in the predicted gates of the policy network. The optimal gate may not be within the range of the highest gate preference, so the search scope needs to be expanded. It can be seen from Figure 10b,d,f that, when *GP* = 10, the result is closer to the optimal solution.

The gate assignment problem addressed in this paper is integer. Some other metaheuristic algorithms are also commonly used to solve such problems [41]. In order to verify the efficiency of the proposed IL-GA algorithm, this paper selects the basic ant colony optimization algorithm (ACO) and particle swarm optimization algorithm (PSO) for comparison. These two metaheuristic algorithms are also used to solve the above eight instances and are compared in terms of the average computing time, the average number of iterations required to reach the optimal solution, and the average optimal value. Figure 11 shows comparison results of these three algorithms. For the gate assignment model established in this paper, the average computing time of PSO is 36.5 s, the average number of iterations is 148.2, and the average flight rate allocated to the contact gate is 68.4%; the average computing time of the ACO is 48.7 s, the average number of iterations is 122.4, and the average flight rate allocated to the contact gate is 65.3%; the average computing time of the IL-GA algorithm proposed in this paper is 20.8 s, the average iteration number is 65.8, and the average flight rate allocated to the contact gate is 81.5%%. It can be seen that the IL-GA algorithm proposed in this article can achieve better optimization results than the other two algorithms in a shorter computational time. With the help of imitation learning, the genetic algorithm starts from a better initial solution, and although it has fewer iterations, its local search efficiency is higher. This further proves the performance of the IL-GA algorithm.

Furthermore, we considered analyzing the rationality of the allocation scheme. Taking instance 8 as an example, Figure 12a shows the Gantt chart representation of the manual assignment result of this instance. The horizontal axis in the Gantt chart represents the time, the vertical axis represents the gate number, and a bar represents a flight pair. In the manual assignment result, only 47 flights are allocated to the contact gates. After using the algorithm proposed in this paper, the assignment result is shown in Figure 12b. There are 58 flights allocated to the contact gates, and the flight rate assigned to contact gates increased by 14.9%. What is more noteworthy is that 22 of these flights were allocated to the same gates as the manual result and 60 flights were allocated to the top five gates predicted by the policy network. Gates 4, 20, 21, 22, and 28 were not used in the manual allocation, indicating that the preference of these gates is low. After imitation learning, our policy network did not allocate flights to these gates either. This shows that the hybrid genetic algorithm based on imitation learning proposed in this paper can not only learn the existing allocation mode and preference of the airport and recommend frequently used gates for each flight from a micro perspective but also help the airport to improve the flight rate allocated to contact gates from a macro perspective. Regarding the airport, staff would be more willing to use a scheme that is similar to their existing methods and has relatively better results than using a new but unfamiliar optimal scheme. Therefore, the method proposed in this paper is more practical in cases where a fully automatic gate allocation system has not been realized.

## 6. Conclusions

In this paper, a hybrid genetic algorithm based on imitation learning is proposed in order to solve the airport gate assignment problem. The goal is to improve the number of flights allocated to contact gates and the total gate preferences. The innovation of the algorithm lies in the use of deep neural networks to learn the trajectory of human gate assignment experts and guide the search direction of the algorithm. The mathematical gate assignment model is transformed to a Markov decision process. The state space, action space, reward, and state transition are defined. In the first stage of the algorithm, the expert assignment trajectory is converted into training samples and labels, and a policy network that can output the gate selection probability of the flights is trained. In the second stage, the policy network is used to generate a good initial population and a genetic algorithm is used for further optimization. The experimental results show that the hybrid algorithm proposed in this paper has a more efficient search process and that the flight rate allocated to the contact gates is 14.9% higher than the manual allocation result and 4% higher than the traditional genetic algorithm. In addition, learning the expert allocation of data also makes the allocation scheme more consistent with the existing allocation methods and preferences of the airport, which is helpful for the practical application of the algorithm. The imitation learning used in this paper is a supervised learning method, so future research could consider using reinforcement learning methods to obtain a more powerful gate allocation decision model and truly achieve an end-to-end solution using deep neural networks.

## Figures and Tables

**Figure 1 entropy-25-00565-f001:**
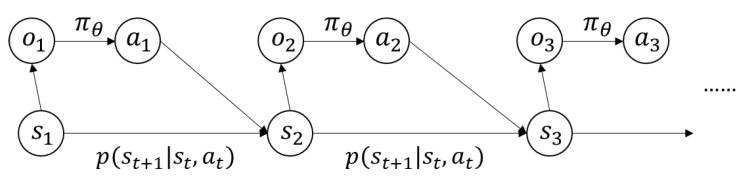
The mechanism of the Markov decision process.

**Figure 2 entropy-25-00565-f002:**
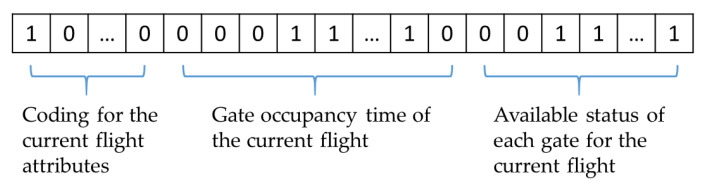
An example of state representation for AGAP.

**Figure 3 entropy-25-00565-f003:**
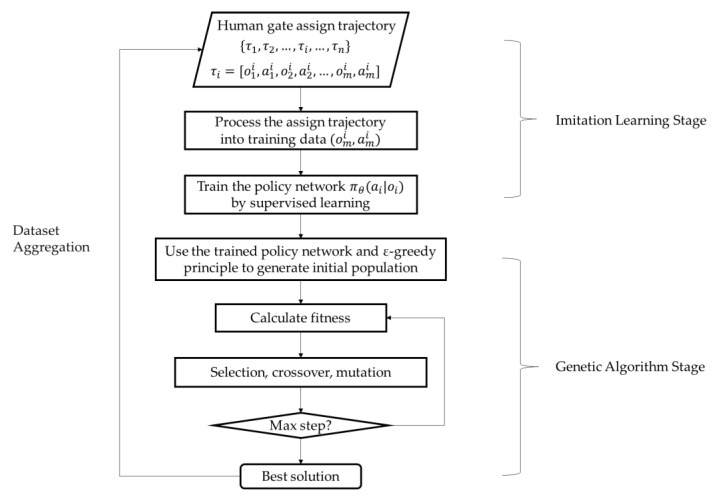
A schematic diagram of IL-GA for AGAP.

**Figure 4 entropy-25-00565-f004:**
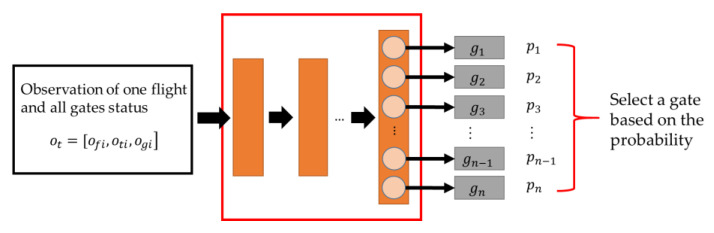
The structure of policy network for imitation learning.

**Figure 5 entropy-25-00565-f005:**
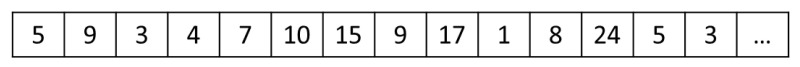
The chromosome structure design of AGAP.

**Figure 6 entropy-25-00565-f006:**
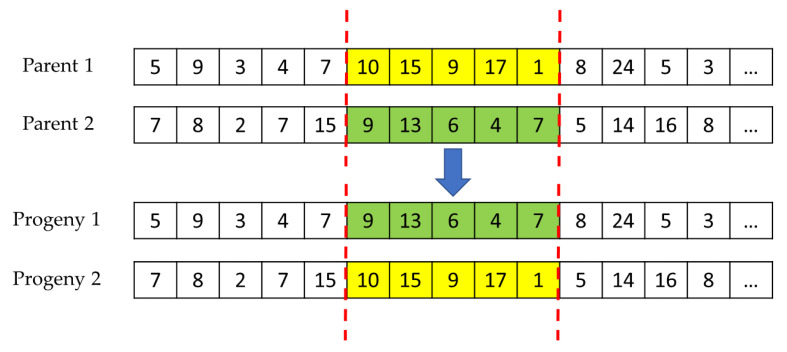
The process of the crossover operator.

**Figure 7 entropy-25-00565-f007:**
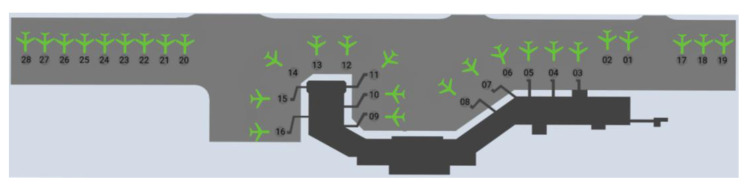
Gate distribution of Lijiang Airport in Yunnan, China.

**Figure 8 entropy-25-00565-f008:**
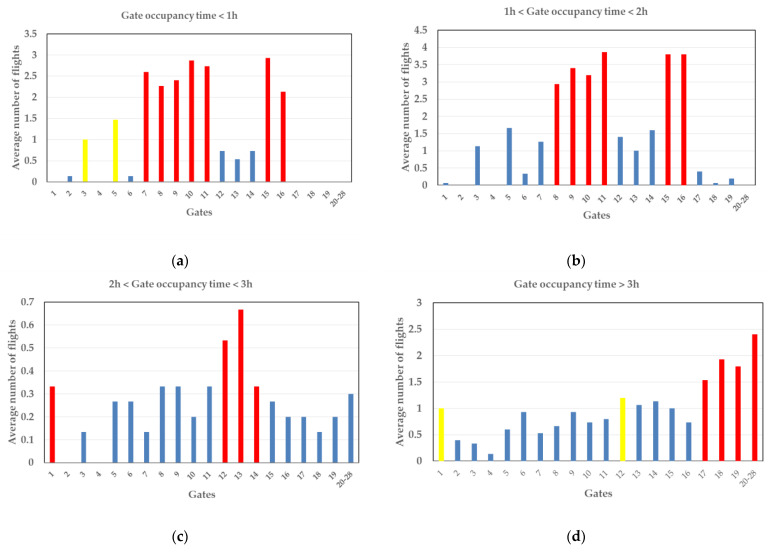
The influence of flights with different occupancy times on the selection preferences of gates (Red gates have a higher preference, followed by yellow, and the rest are blue): (**a**) for flights with an occupancy time of less than 1 h, gates 15, 16, 11, 10, and 9 are mainly chosen; (**b**) for flights with an occupancy time of 1–2 h, gates 15, 16, 11, 10, 9, and 8 are mainly chosen; (**c**) for flights with occupancy time of 2–3 h, gates 12, 13, 14, and 1 are mainly chosen; (**d**) for flights with an occupancy time more than 3 h, gates 17, 18, 19, and 20–28 are mainly chosen.

**Figure 9 entropy-25-00565-f009:**
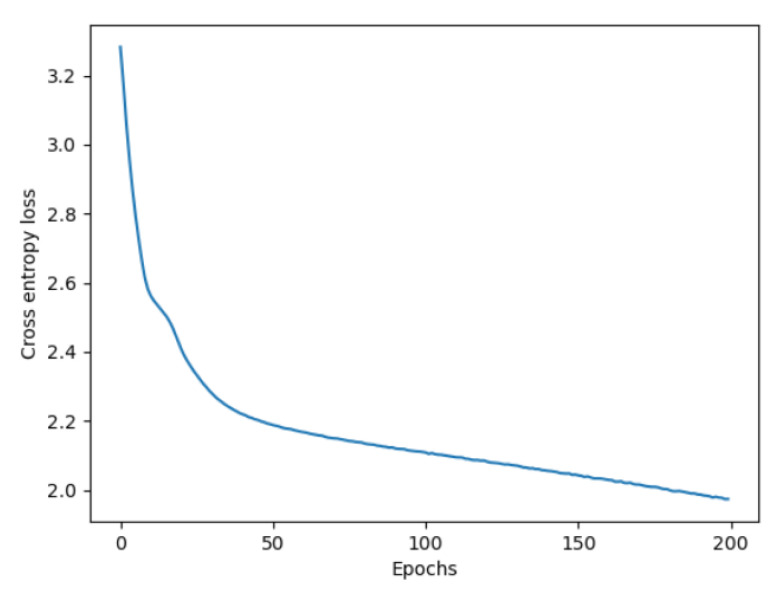
The iterative process of cross-entropy loss in the training epochs.

**Figure 10 entropy-25-00565-f010:**
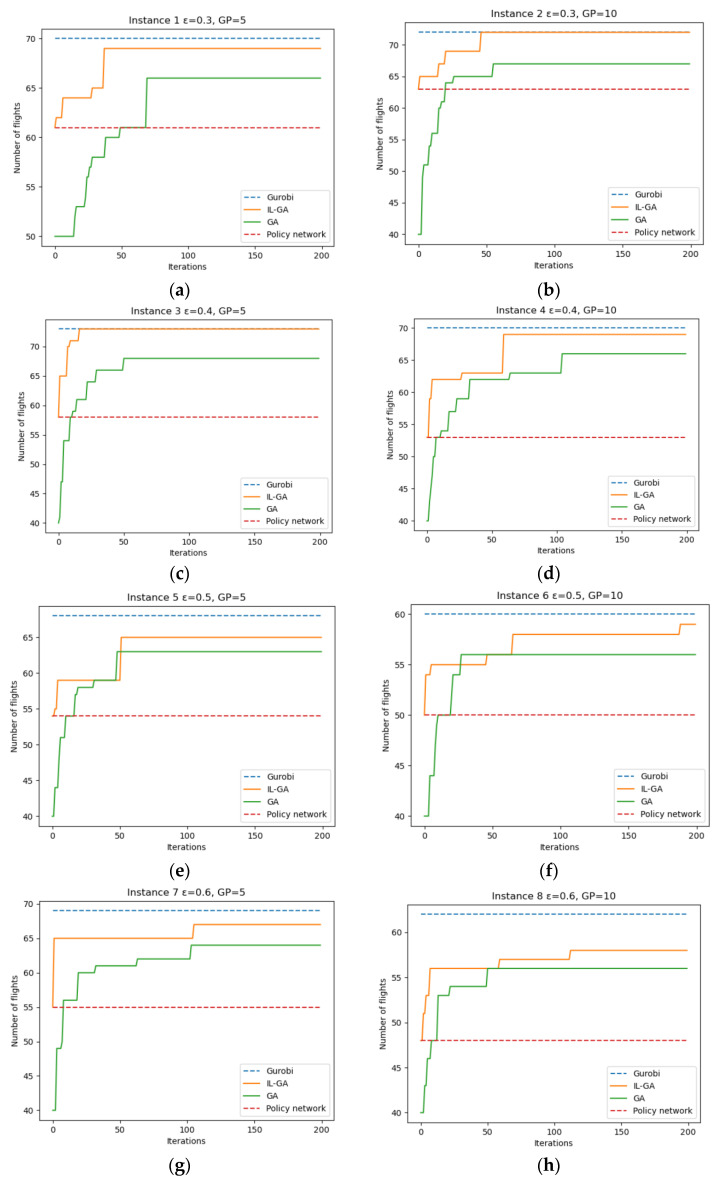
A comparison of the iterative process for each instance: (**a**) instance 1 with parameters ε=0.3 and GP=5; (**b**) instance 2 with parameters ε=0.3 and GP=10; (**c**) instance 3 with parameters ε=0.4 and GP=5; (**d**) instance 4 with parameters ε=0.4 and GP=10; (**e**) instance 5 with parameters ε=0.5 and GP=5; (**f**) instance 6 with parameters ε=0.5 and GP=10; (**g**) instance 7 with parameters ε=0.6 and GP=5; (**h**) instance 8 with parameters ε=0.3 and GP=10.

**Figure 11 entropy-25-00565-f011:**
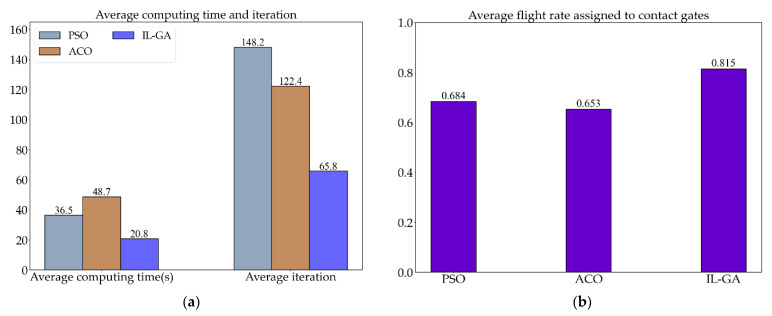
The comparison result between PSO, ACO, and IL-GA: (**a**) the average computing time and iteration; (**b**) the average flight rate assigned to contact gate (optimal value).

**Figure 12 entropy-25-00565-f012:**
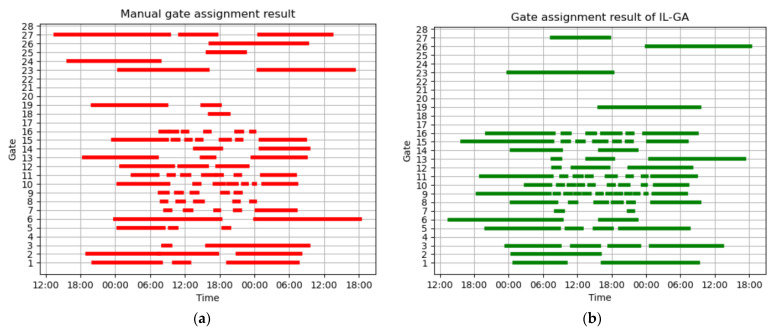
The comparison between manual gate assignment result and IL-GA gate assignment result of instance 8: (**a**) the manual assignment result; (**b**) the assignment result of the proposed IL-GA algorithm.

**Table 1 entropy-25-00565-t001:** Parameters and decision variables used in the gate assignment model.

Parameters	
F	The set of flight pairs, the total number is |F|
G	The set of gates, the total number is |G|
${TA}_{i}$	The arrival time of flight pair i, i∈F
${TD}_{i}$	The departure time of flight pair i, i∈F
$f_{i}^{T}$	The aircraft type of flight pair i
$f_{i}^{N}$	The international/domestic type of flight pair i
$f_{i}^{V}$	The VIP type of flight pair i
$f_{i}^{O}$	The overnight type of flight pair i
$g_{k}^{N}$	The international/domestic type of gate k, k∈G
$g_{k}^{B}$	Whether gate k is a contact gate
$g_{k}^{T}$	Types of aircraft allowed to park in gate k
$p_{ik}$	The preference score for flight pair i and gate k
$T_{buffer}$	The minimum safety time interval of the same gate
$T_{neighbor}$	The minimum safety time interval between two neighbor gates
**Decision variable**	
$x_{ik}$	0–1 decision variable, $x_{ik}=\left\{\begin{array}{c} 1,\;\mathrm{flight}\;i\;\mathrm{is}\;\mathrm{assigned}\;\mathrm{to}\;\mathrm{gate}\;k\\ 0,\;\mathrm{flight}\;i\;\mathrm{is}\;\mathrm{not}\;\mathrm{assigned}\;\mathrm{to}\;\mathrm{gate}\;k \end{array}\right.i\in F,\;k\in G$

**Table 2 entropy-25-00565-t002:** Attributes of each gate.

Gate. No	Gate Type	Nation Type	Aircraft Types
1~2	remote gate	domestic	B733-B738/A319/A320
3	contact gate	international	B733-B738/A319/A320
4	contact gate	domestic	Business aircraft
5	contact gate	international	B733-B738/A319/A320
6	remote gate	domestic	B733-B738/A319/A320
7	contact gate	domestic	B737
8–10	contact gate	domestic	B733-B738/A319/A320
12–14	remote gate	domestic	B733-B738/A319/A320/A321/B763
15–16	contact gate	domestic	B733-B738/A319/A320
17–19	remote gate	domestic/international	B733-B738/A319/A320
20–28	remote gate	domestic	B733-B738/A319/A320

**Table 3 entropy-25-00565-t003:** Gate assignment trajectories from manual experience.

Flight. No	Aircraft Type	Arrival Time	Departure Time	VIP	Overnight	Nation	Assigned Gate
1	A319	20:01	06:55 (+1)	N	Y	D	14
2	A320	23:31	09:47 (+1)	N	Y	D	3
3	A320	23:48	07:42 (+1)	N	Y	D	10
4	A319	00:12	09:10	N	Y	D	17
5	B737	00:43	08:54	N	Y	D	7
6	B737	07:05	11:24	N	N	D	1
7	A320	07:28	08:35	Y	N	D	15
8	B737	08:13	09:20	N	N	D	11
9	B737	09:22	10:29	N	N	D	9
10	A320	09:56	11:39	N	N	D	12
11	A320	10:17	11:28	N	N	I	3
12	A320	10:33	14:20	N	N	D	5
…	…	…	…	…	…	...	…
14,000	A320	15:07	07:16 (+1)	N	Y	D	25

**Table 4 entropy-25-00565-t004:** Structure design of the policy network.

Layer	Settings
Input layer	Number of units = 313
Hidden layer 1	Number of units = 512, activation function = ReLU
Hidden layer 2	Number of units = 512, activation function = ReLU
Output layer	Number of units = 28
Softmax layer	Number of units = 28

**Table 5 entropy-25-00565-t005:** The training hyperparameter settings of policy network.

Parameter	Value
Epochs	200
Batch size	64
Learning rate	0.01

**Table 6 entropy-25-00565-t006:** Comparisons between the predictive gates and actual gate of flights.

Flight. No	Arrive Time	Departure Time	Predict Gates	Actual Gate
1	15:07	07:16 (+1)	(25, 24, 27, 19, 1)	25
2	20:04	08:02 (+1)	(2, 1, 17, 14, 13)	2
3	20:10	07:01 (+1)	(14, 13, 17, 12, 2)	17
4	07:15	08:36	(16, 8, 7, 15, 5)	5
5	07:52	09:05	(8, 11, 10, 16, 15)	8
6	08:08	10:05	(11, 15, 9, 12, 14)	12
7	08:34	09:41	(9, 8, 15, 16, 11)	15
8	09:04	11:36	(17, 12, 6, 1, 2)	2
9	09:13	10:20	(16, 10, 15, 9, 11)	11
10	09:35	10:42	(5, 10, 8, 16, 9)	9
…	…	…	…	…

**Table 7 entropy-25-00565-t007:** The parameter settings of genetic algorithm for AGAP.

Parameter	Value
Population size N	200
Iterations t	200
Greedy principle ε	0.3~0.6
Number of gates selected by the policy network GP	5, 10
Minimum safety time interval of the same gate $T_{buffer}$	10 min
Minimum safety time interval between two neighbor gates $T_{neighbor}$	5 min

**Table 8 entropy-25-00565-t008:** The parameter settings and optimal result statistics of the instances.

Instance	Flights	*ε*	*GP*	Flights Assigned to Contact Gate			
				Gurobi	IL-GA	GA	Policy Network
1	81	0.3	5	70	69	66	61
2	88	0.3	10	72	72	67	63
3	85	0.4	5	73	73	68	58
4	82	0.4	10	70	69	66	53
5	84	0.5	5	68	65	63	54
6	75	0.5	10	60	59	56	50
7	83	0.6	5	69	67	64	55
8	74	0.6	10	62	58	56	48
**Average flight rate assigned to contact gates**				83.4%	81.5%	77.5%	67.7%

## Data Availability

The data presented in this study are available on request from the corresponding author. The data are not publicly available due to the confidentiality requirements of the airport flight data.

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
