# Peer review of "A Hybrid Genetic Algorithm Based on Imitation Learning for the Airport Gate Assignment Problem"

_entropy, 2023, doi:10.3390/e25040565_

Round 1

Reviewer 1 Report

I do not have any additional recomendations. I suggest to publish the paper without any revisions.

Author Response

Thanks for taking your time to review our article. We are grateful for your suggestion to publish our manuscript.

Reviewer 2 Report

1. Interesiting and well written paper focusing on an important practical problem: Airport Gate Assignment Problem.
2. Although the proposed hybrid algorithm is based on two known paradigms: genetic algorithm and impitation learning, their hybridization is original and, as computational experiment proved, efficient.
3. The paper is written in clear and consistent form.
4. Scientific contribution is stated clear.
5. Literature review part refers to the papers presenting state-of-the-art methods and algorithms.
6. Presentation of the proposed approach is clear and correct. I would only suggest to write algorithms or parts of algorithms in similar way as Algorithm 1 (page 11).
7. Computational experiment has been carried out correctly. Conclusions sound clear.

I suggest to accept the paper.

Author Response

Thanks for your positive comments and feedback on our manuscript.

The Algorithm 1 in page 11 is the core of the algorithm proposed in this paper, so we focus on the description of its steps.  Other parts of the algorithm pay more attention to the specific design of some operators, so we think it is clearer to express them by pictures than the steps, such as Figures 4 to 6.  Of course, we also appreciate your suggestions.

Reviewer 3 Report

The manuscript proposes the use of a IL-GA method to solve the Airport Gate Assignment Problem.

Some remarks:

1- The 2nd paragraph of Section 2 is too long. Please, split it;

2- Section 3 lacks references. Many strong statements we done without the properly citation;

3- The same for Section 4;

4- "through selection, crossover and mutation with the help of genetic operators of natural genetics."

-> crossover and mutation are the genetic operators;

5- It is widely knwon the roulette wheel method tends to premature convergence. Why the authors did not use binary tournament? 

6- The problem addressed is integer. The literature is convergent in show that the best bio-inspired model to deal with this kind of task is the ACO. The authors should consider this possibility;

8- In the same way, other possibilities should be considered, as PSO. The following manuscript can be useful: 10.1109/ACCESS.2021.3124710;

7- Table 7: the variables must be italics. 

Author Response

Thank you for reviewing our manuscript and giving us some remarks. We have revised it based on your comments. Please see the attachment for the response to each remark.

Round 2

Reviewer 3 Report

The authors have modified the manuscript as I suggest. The only minor correction is regarding the following: " three genetic operators: selection, crossover, and mutation". The selection is not a genetic operator. So, there are only two genetic operators, crossover and mutation. Please, correct this small error. Congratualtions!